# A Study on the Characteristics of New Towns and the Redevelopment of Project-Canceled Areas: A Case Study of Seoul, South Korea

Hyunjung Lee

Associate Research Fellow, Department Urban Planning and Design Research, the Seoul Institute, 57 Nambusunhwan-ro, 340-gil, Seocho-gu 06756, Korea; bb0028@hanmail.net

**Abstract:** This study examined regeneration issues from the perspective of urban regeneration and the characteristics of areas that have been the subject of public policy failure, namely new towns and redevelopment of project-canceled areas. These areas are in need of improvement, particularly in regard to old housing and poor infrastructure. It is imperative that infrastructure conditions that may be difficult to focus on in the private sector are improved. Therefore, the public (Seoul Metropolitan Government (SMG)) needs to play an active role, with a particular focus on providing significant administrative power and finances to these areas in order to reduce the effects of policy failure and make continuous efforts to reverse the failed policy. The public should actively work to resolve distrust and conflict in the public policy and make restorative efforts through new policies.

**Keywords:** new towns and redevelopment of project-canceled areas; canceled areas; urban regeneration; failure of policies; residential area decline

## 1. Introduction

### 1.1. Background and Purpose

Urban regeneration can be seen as part of the effort to cope with the decline of existing urban industries, population outflow, and aging housing. Urban regeneration promotes industrial, economic, and sociocultural activities as well as the physical environment. It is possible to achieve this by introducing and reinforcing new functions in order to revitalize the competition that has arisen from this decline in the urban environment, which is now stagnant [1]. The discussion regarding urban regeneration began in the 1980s in the UK to cope with the economic downturn and the decline of cities due to the rapid changes in industrial structure [2].

Many cities around the world are pursuing various policies and projects regarding urban regeneration, with diverse methods and characteristics. Over the past 20 years, countries around the world have focused on urban regeneration through the redevelopment of public or private real estate. Recently, however, an emphasis on more diverse issues, such as the urban regeneration process and cooperation among subjects, has become apparent [3]. Although various urban development projects have been promoted to overcome city decline throughout the world recently, despite the expectation that real estate development on an appropriate scale would contribute to urban regeneration, there were many concerns about market-led development: for example, the expansion of the economic gap in cities and the deterioration of residents' quality of life [4]. In particular, traditional Asian cities, including cities in China, are arguing that positive social goals should be considered in regard to the urban development process by questioning the sustainability of regions undergoing rapid urban change and neighborhood dismantlement as a result of large-scale redevelopment [5,6].

Seoul experienced rapid urban growth from the 1970s to the 1980s and the 1990s, and the decline of urban areas greatly increased in the 1990s. This was due to the continuous movement of populations, housing, businesses, etc., from downtown to the Gangnam, which is represented by high housing and land prices due to its location in south Seoul, its pleasant residential environment, and its active commercial area [7]. In order to cope with this decline in the city center, preservation of the historical downtown area and revitalization policies have been promoted since the early 2000s, with the downtown urban regeneration project most representative of this focus being the restoration of Cheonggyecheon (the area is a 10.9 kilometer long (6.8 mi), modern public recreation space in downtown Seoul. The massive urban renewal project is on the site of a stream that flowed before the rapid post-war economic development caused it to be covered by transportation infrastructure. The $900 million project initially attracted much public criticism but, since opening in 2005, has become popular among residents and tourists) in Seoul. With this decline in the city center, an overall policy focusing on residential area decline was implemented from the 2000s, named the "new town project", in an attempt to develop and regenerate a full-scale residential area.

The main purpose of the new town project was to promote full-scale redevelopment of the demolished area. As problems were revealed, such as a surge in housing prices and a decline in business feasibility due to overestimation of public affiliation, promoting the project became more difficult to achieve while still maintaining the redevelopment goals. The living environment of the residential areas was continuously mistreated and eventually neglected due to project delays [7,8].

Park Won-Soon, the mayor of the Seoul Metropolitan Government (SMG), began an exit plan for Seoul's new town and redevelopment project in 2012 with the help of his municipal government [7], in which the city retracted the failure of the policy and tried to solve the situation. From this point of view, the policy of development and management of residential areas in Seoul changed from front demolition redevelopment to an alternative objective of urban regeneration. The areas designated for the project were released according to the exit plan [7]. The canceled areas were identified in order to recognize and fix the failure of the public implementation policy. The SMG turned its policy direction toward urban regeneration promotion and management.

This study examined the characteristics and regional regeneration issues of canceled areas, which is an area of public policy failure. The Seoul Metropolitan Government implemented the new town project to solve residential area decline, but the project actually played a role in accelerating the decline of residential areas. This study analyzed the characteristics and issues with regard to improving the management of urban regeneration. The purpose of this study was to establish the direction that policy should take based on an accurate understanding of the current situation in these areas.

## 1.2. Data and Methods

This study was based on the spatial scope of Seoul's new towns and the redevelopment of canceled areas. Of the 686 districts surveyed regarding proposed improvement, 393 were canceled until February 2018. An analysis of the conditions of the canceled areas was conducted using public data on physical, economic, and sociocultural characteristics. The field survey and analysis of the geographic information system (GIS) used basic data and region examples.

Data were sourced from management cards of the canceled areas, the building management ledger of Seoul, and the urban planning information system (UPIS), which carries internal administrative data for Seoul.

Physical characteristics analyzed included housing conditions, such as housing construction periods and construction of houses using building ledgers, contact with the road, and parking status, which were combined to analyze the infrastructure status using UPIS data.

Economic and industrial characteristics were analyzed using land prices and housing sales. Sociocultural characteristics were analyzed using the Seoul Metropolitan Government's demographic data on the elderly population, which allowed for a review of the policy consideration target.

*1.3. Structure of the Research*

This study was aimed at analyzing the characteristics of the area from the new town redevelopment project that was canceled.

First, I reviewed the institutional system, such as the object and procedure of the new town redevelopment. Next, the meaning and value of the area as a residential area in Seoul were examined. The policy meaning and the task of urban regeneration of the cancellation area were reviewed. To examine the tasks of urban regeneration, I analyzed the locational, physical, residential environment, economic, and sociocultural characteristics of the entire canceled area in Seoul. Based on the analysis results, issues for regeneration of the canceled area were derived. In order to closely review the need for regeneration of public policy failure, the actual case study was more concretely specified, focusing on the case of Sajik 2 district. Through this, the issue and direction of regeneration policy on the public policy in Seoul was examined from the urban regeneration perspective (Figure 1).

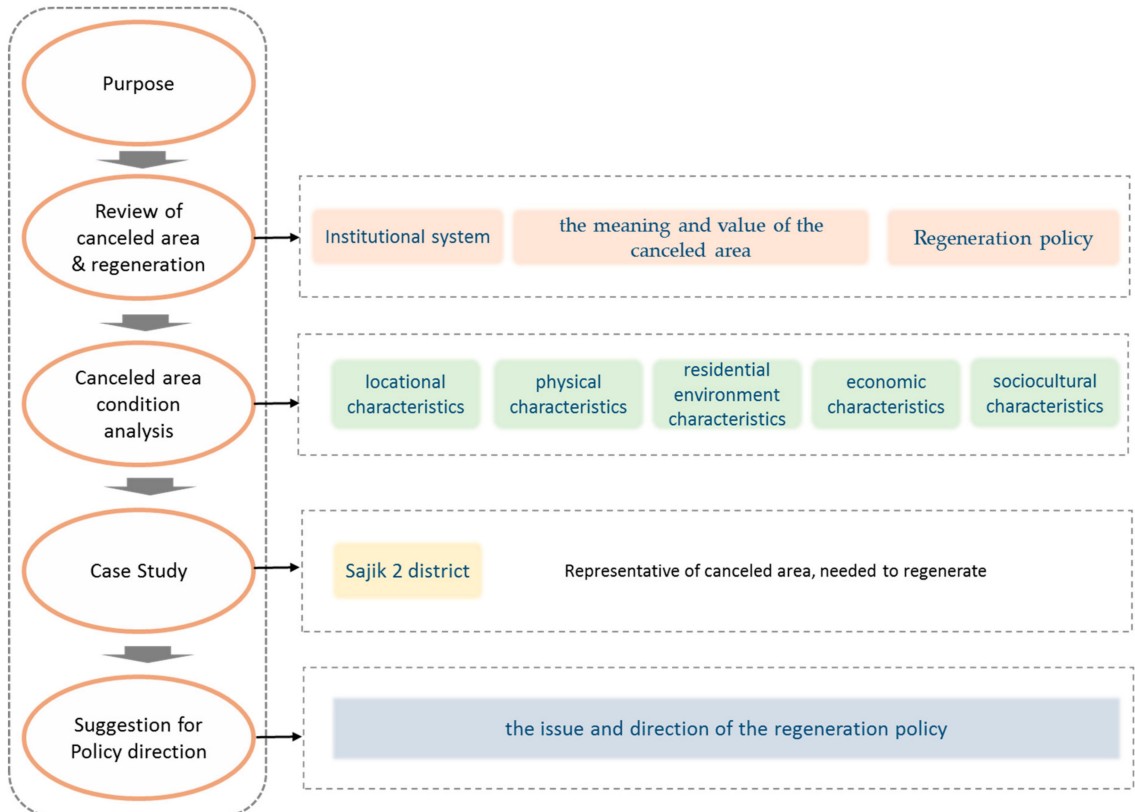

**Figure 1.** Structure of the research.

## 2. Relationship between Canceled Areas and Urban Regeneration

*2.1. Targets and Procedures for Canceled Areas*

Areas are canceled by the request of the owners, the sunset system (a system that automatically eliminates the effects of laws and regulations over a certain period of time, just as the sun goes down over time), the establishment of the promotion committee, and the union (landowners' union for housing redevelopment projects), in accordance with Article 4 of the Act on the Improvement of Urban Areas and Residential Environments (cancellation of improvement of the district, etc.) and Article 16-2 (cancellation of the union establishment permit) (Table 1) [7–9].

**Table 1.** Requirements and procedures for improvement (proposed improvement) of districts [10].

| | Requirements | Procedures |
|---|---|---|
| Sunset system | -Three years from scheduled date of improvement district designation if improvement district is not specified. -If committee approval is not applied for within two years after designating an improvement district. -If union establishment is not applied for within two years after committee is formed. -If application for execution improvement is not applied for within three years after establishment of union. | Released after public notice of 30 days, opinion gathering from local councils, and deliberation by Urban Planning Commission. |
| Deactivation of authority | -Cost-sharing, in which land owners exceed what is expected of them. -If owner requests issue of cancellation of 30% of land, etc., in improvement district, in which planned improvement area or propulsion committee is not formed. | Lifted after deliberation by Urban Planning Commission. |
| Cancellation of establishment of promotion committee and union | -If about half to two-thirds of consent is obtained from propellant and union establishment or owner gives half of consent, improvement district is deregulated. | Lifted after deliberation by Urban Planning Commission. |

This study dealt with areas that were designated for improvement but were then canceled according to the SMG's actual condition survey assessing districts designated for proposed improvement. This survey was based on the Act on the Improvement of Urban Areas and Residential Environments and the Ordinance.

The SMG also conducted a survey that assessed the actual conditions of districts planned for improvement according to the exit plan for Seoul's new town and redevelopment project [7]. The survey and an estimation regarding the project's feasibility were conducted regarding the districts designated for improvement (proposed improvement) without the redevelopment project association. The results were provided to the residents (landowners, etc.) who were part of determining whether to cancel or promote the proposed improvement of their district. Based on the results of the survey, more than 30% of landowners requested that the proposed improvement district be canceled and took steps to cancel the improvement project [7].

Since the SMG promoted cancellation of the planned improvement district in 2012, 393 of the 686 survey subjects were released as of February 2018. Alongside this, 262 sites were designated to continue with the improvement project and 28 sites were still undetermined regarding cancellation (Figure 2) [11,12].

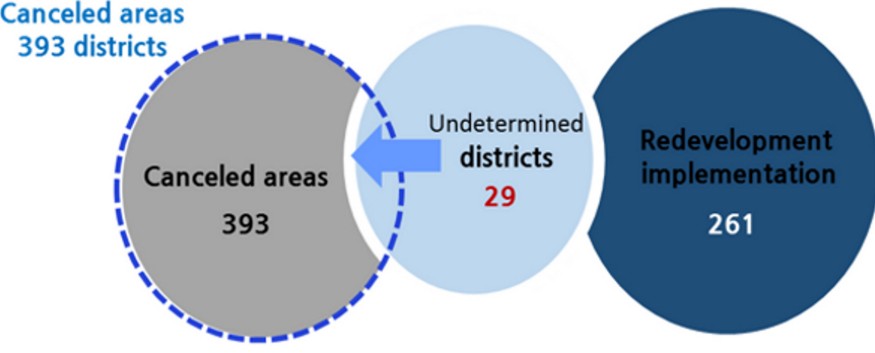

**Figure 2.** Proposed improvement districts and implementation progress [11].

The low-rise residential areas in Seoul can be divided into separate improvement districts, including an improvement project implementation area, an improvement project release area, an urban regeneration project area, and a general low-rise residential area where implementing improvement is not applicable [12].

Although it is necessary to improve deteriorating local environments, such as those present in canceled areas, if there is no current effective alternative, various plans for management of the area are needed. The general residential areas need to be planned in accordance with the zoning system and other comprehensive plans (Figure 3) [11].

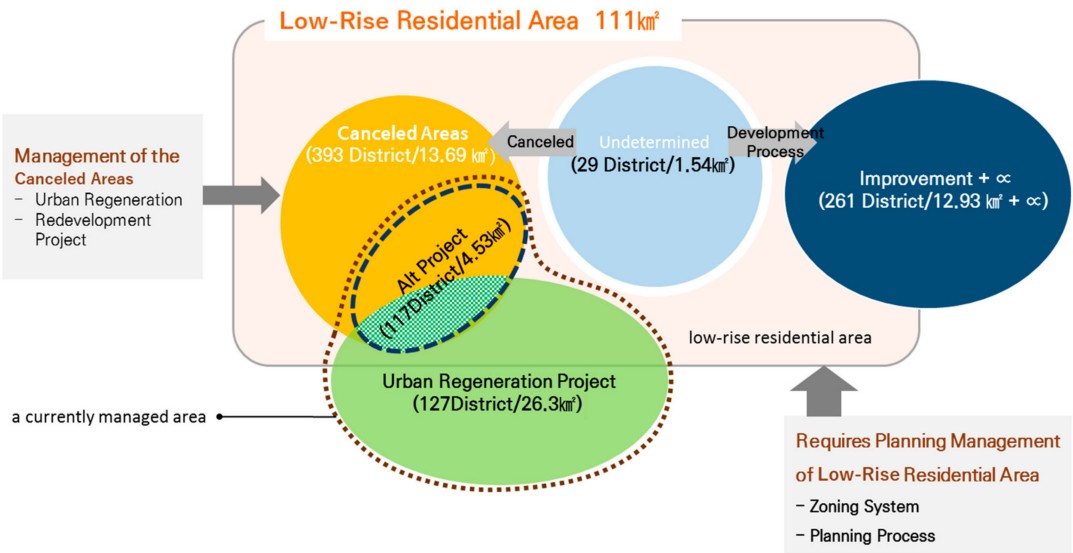

**Figure 3.** Relationship between canceled area and Low-Rise residential area [11].

## 2.2. Status of Canceled Areas in Low-Rise Residential Areas

The residential areas in Seoul can be divided into low-rise areas with fewer than four or five floors, and high-rise areas, such as apartments. Environmental improvement in the low-rise residential areas was carried out by demolition redevelopment, and the environment of high-rise residential areas was improved through reconstruction projects.

However, this redevelopment project has become more difficult to operate due to changes in social and economic conditions in the low-rise residential areas; therefore, the necessity of regeneration projects, not redevelopment of the entire demolition, has increased. Low-rise residential areas provide relatively cheap and diverse housing and play a big role in many citizens' lives [9]. Most were formed before the 1990s and changes have mainly occurred via individual plot developments, showing the appearance of single-family and multi-family houses in residential areas. The low-rise residential zone has an area of about 111 km$^2$ and accounts for about 35% of the city's developed area (Figures 4 and 5) [13,14].

The SMG implemented the Urban Regeneration Project for the planned management and maintenance of low-rise residential areas. This was a comprehensive plan including development, improvement, conservation, and economic and cultural welfare in response to sustainable development, low growth, and important policies for administrative and private cooperation efforts [15]. The SMG made a large effort to revitalize the city through residential regeneration of canceled areas. The canceled areas were regions where there was a need to improve the physical environment: for example, poor residential environments due to aging housing facilities and a lack of infrastructure such as parking lots and public facilities. According to administrative procedure, residents who expected to build apartments through new, large-scale development felt a great sense of loss, increasing distrust in

the SMG. Redeveloping the entire demolition area was considered to be the only solution to improve housing and infrastructure, but it was canceled, causing public–private conflict.

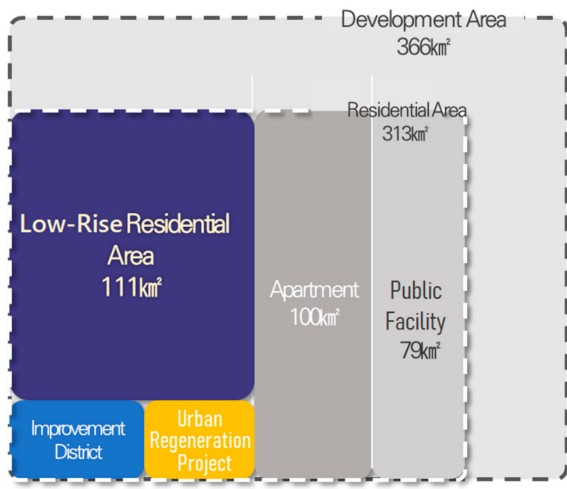

**Figure 4.** Residential areas in Seoul [11].

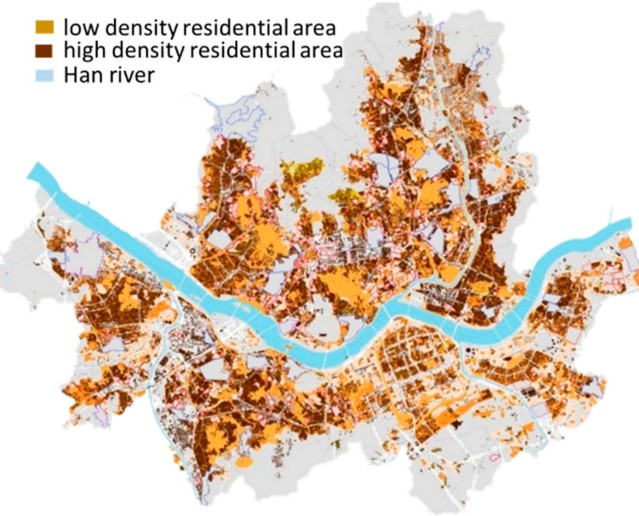

**Figure 5.** Low-rise residential area map.

*2.3. Policy Meaning and Urban Regeneration of Canceled Areas*

For the past half-century, Seoul has experienced rapid urban growth as a result of quick economic growth. Since 2000, the existing development area has declined as the area has aged; as a result, efforts have been made to improve the local environment through redevelopment projects (Figure 6) [16]. The most representative case of a residential area where such effort was made is the Seoul new town project plan [10].

The new town project started as part of the regional balanced development plan for underdeveloped areas of Seoul over a four-year period (2002–2006). In December 2002, three districts, Eunpyeong, Gilum, and Wangsimni, were selected as pilots for the project to plan redevelopment of poor, aging housing areas in the Gangbuk area (the city area north of the Han River). A total area of 23.8 km$^2$ was later designated in 2003 through the addition of 12 designations for the second new town project (8 in Gangbuk and 5 in the southwest–southeast), and the third project involved 11 districts, and was carried out around 2007. This was a wide area, about 2.4 times the area of the 10.1 km$^2$ used for about 30 years from 1973 to 2003 (Table 2) [7,10].

**Table 2.** New town project designation in Seoul [10].

|  | District | Area (km²) | Population | No. of Households |
|---|---|---|---|---|
| Pilot new town project | 3 | 5.1 | 97,745 | 35,478 |
| Second new town project | 12 | 8.2 | 366,927 | 153,735 |
| Third new town project | 11 | 10.5 | 390,237 | 158,480 |
| Sum | 26 | 23.8 | 854,909 | 347,693 |

The new town project was an apartment reconstruction project aimed at improving the residential environment in declining districts, supply infrastructure, and restore urban functions. It was a project plan to improve housing and secure infrastructure in a mega-plan site. The development of the mega-plan site was designed to upturn zoning regulations and reduce the floor area ratio, which caused concerns regarding overloading the development during the project implementation process. The project was overdesignated as a pledge to appoint a new town by politicians during the election, and due to the surge in real estate prices from 2002 to 2007, which caused a blow-up of the real estate market, the new town project ended up causing many problems [10].

In 2008, the housing market rapidly cooled down due to the global financial crisis; housing prices fell and the burden on owners increased significantly, resulting in the expansion of the new town project being halted, and then stopping without further progress. As project progress stopped and the real estate market deteriorated, the SMG was confused about urban management due to problems such as public complaints and accelerated decline. In order to solve this problem, the administration tried to change the model into urban regeneration as an alternative improvement. This was not large-scale development, but rather a reflection of the failed administrative policy [15]. Park Won-Soon, the mayor of the SMG, announced the exit plan for Seoul's new town and redevelopment project and released the proposed improvement for districts where development progress was no longer possible due to low business feasibility [7,17,18].

This trend continued until 2019, with various types of urban regeneration projects being implemented for the canceled areas. Seoul is currently actively promoting regeneration projects for urban areas, including residential areas, through the excavation and promotion of a "Seoul-type urban regeneration project" [19–23].

The most important impact of the canceled areas can be found in the public's continuous efforts to reflect on and reverse the failed policy by providing huge administrative power and finance. In a city that is reborn and experiences the processes of growth and decline continuously, this situation is an example of how long it takes for a wrong policy decision to become established and progress further with regard to urban policy, and how much budget and administrative power must be provided [11,24–26].

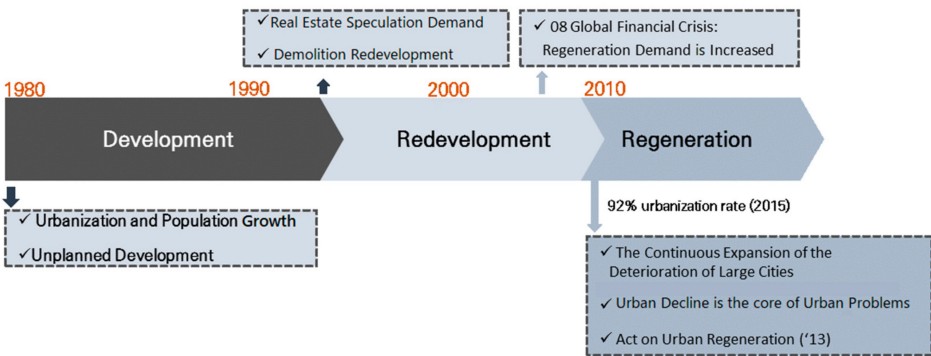

**Figure 6.** City development policy by period.

### 3. Actual Condition of Canceled Areas and Urban Regeneration Issues: A Case Study

As mentioned above, this study analyzed the actual conditions of areas affected by the policy failure through data. This analysis reviewed the characteristics of residential areas that need policy consideration and can serve as a basic research foundation for policy direction in regards to residential area management in Seoul in the future.

*3.1. Basic Status of Canceled Areas*

The 393 canceled areas are about 13.69 km$^2$ in size, accounting for about 12.3% of the total low-rise residential area. This target corresponds to poor, aging housing and infrastructure. There is an urgent need in these areas to improve the quality of the residential environment.

The analysis results of the basic data for these 393 areas were briefly examined in terms of location, physical and residential environments, and socioeconomic aspects.

3.1.1. Location Characteristics

The characteristics of the locations were examined, with a focus on topographical characteristics, public transportation accessibility, and distribution of types of districts.

Most residential areas located in hilly areas tend to be perceived as old and poor housing sites compared to those located in flat areas. This could be because 37.9% of the area is located on hillsides, and the quality of roads and houses in these areas is poorer than that of those located in flat areas.

Generally, when talking about the future development potential and convenience of the region, public transportation conditions are recognized as very important. For this region, about 279 out of 393 areas, or 71.0%, were within a 250 m radius of a subway station or main road; therefore, public transportation is considered to be very good here.

3.1.2. Physical Characteristics

The canceled area was found to have variable characteristics in terms of size and infrastructure in the region, according to the development project and location. In terms of size, there was a big difference in area, from a small area less than 0.1 km$^2$ in size to large area exceeding 1 km$^2$. Areas that were taken from the reconstruction project were generally classified as large areas rather than canceled redevelopment project areas; these areas accounted for 37.7%. The road width was less than 3 m, so the area consisted of narrow roads with very poor vehicle access. In addition, residents stated that there problems in regard to their convenience and access to facilities, such as a lack of parks and parking lots.

3.1.3. Residential Environment Characteristics

Over 36% of the housing, which accounted for the majority of the canceled area, included single-family housing with up to two floors that was older than 30 years, which signifies a high demand for housing improvements (Figures 7 and 8).

Figures 6 and 7 show the ratio of buildings older than 20 or 30 years in the canceled areas; in 87.5% (341 of 393 districts), more than 60% of the total buildings in the area were over 20 years old, and buildings over 30 years old were a majority for 141 districts (Figures 7 and 8).

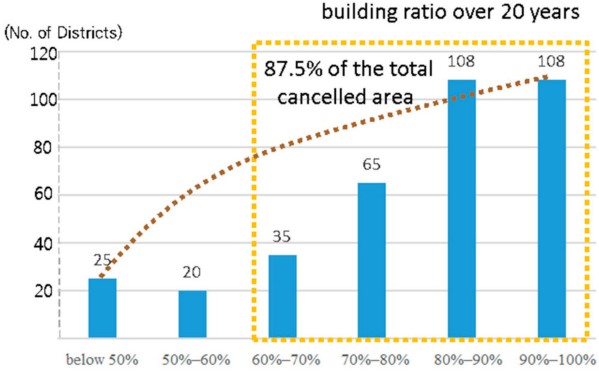

**Figure 7.** Building ratio over 20 years [11].

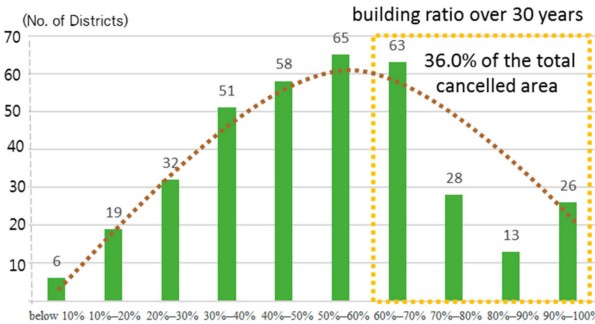

**Figure 8.** Building ratio over 30 years [11].

As a result of analyzing the construction activity in these areas over the last 10 years, it was found that multi-family houses were newly built with relatively good development conditions, and were transformed into dense, multi-family dwellings with fewer than five floors. This type of housing development resulted in new urban problems, such as parking and vehicle congestion, because it was not accompanied by infrastructure improvements corresponding to the increased numbers of household occupants (Figure 9).

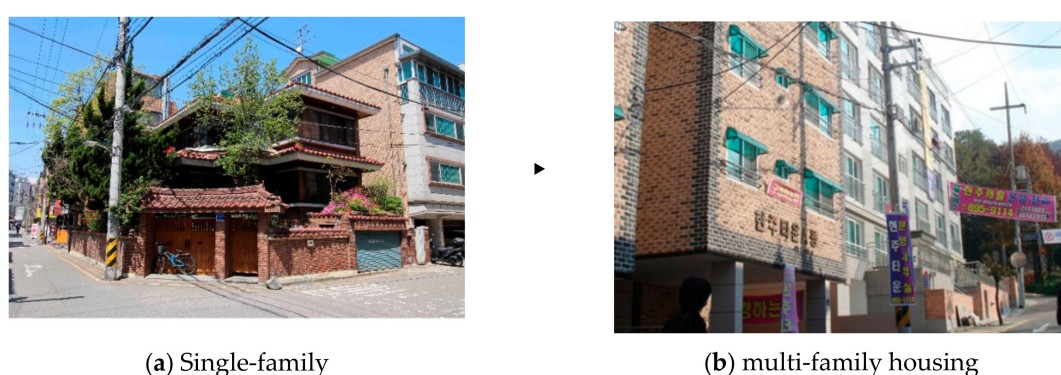

(**a**) Single-family　　　　　　　　　　　　　　　　(**b**) multi-family housing

**Figure 9.** Single-family and multi-family housing.

In some regions of the canceled area, migration due to demolition was already occurring during the project, and an area appeared consisting of collective vacant houses. This was regarded as a serious problem, and policy implementation, such as the purchase of empty houses, was carried out. Although vacant houses were shown to be responsible for serious problems such as accelerated decline in the whole region and security problems, it was difficult to apply double administrative activities, such as application of the redevelopment project, to an area where cancellation was already decided.

This caused a situation in which neither development nor decline was appropriate (Figure 10). In one release area (of Sajik 2 district), 87 of 195 houses, or about 45%, were left empty.

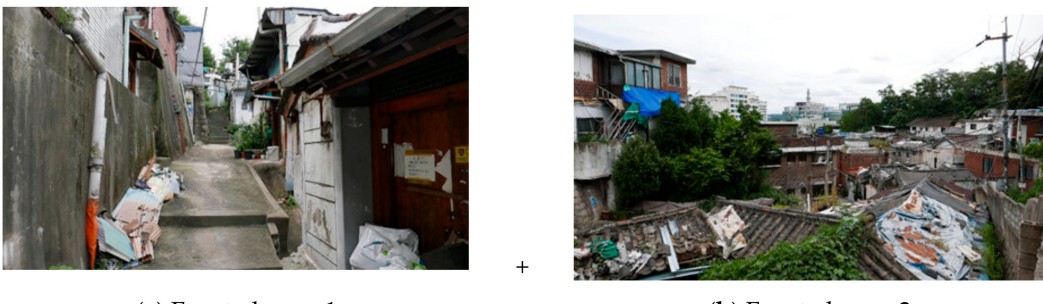

(**a**) Empty house 1          +          (**b**) Empty house 2

**Figure 10.** Empty, unattended houses in the canceled area.

### 3.1.4. Economic Change Characteristics

The land price changes during 2001–2017 were compared with the period of designation and release of the canceled areas. This analysis showed that the increase of land prices slowed after the release. It can be assumed that this was caused by psychological factors, i.e., people did not expect to see a profit for new houses through the redevelopment. Expectations for possibly enjoying development profits were lowered, which also caused conflict between the private (landowners) and public (SMG) sectors (Figure 11). After cancellation of the planned area, a rapid increase in land prices appeared in some places. These regions represented the hot areas that were commercialized by utilizing the image of the old and friendly area in the urban regeneration projects. As mentioned above, multi-family houses tended to be constructed in the canceled areas, with changes in ownership due to land and house sales in the area after the release. The number of sales increased significantly after the release of each region (Figure 12).

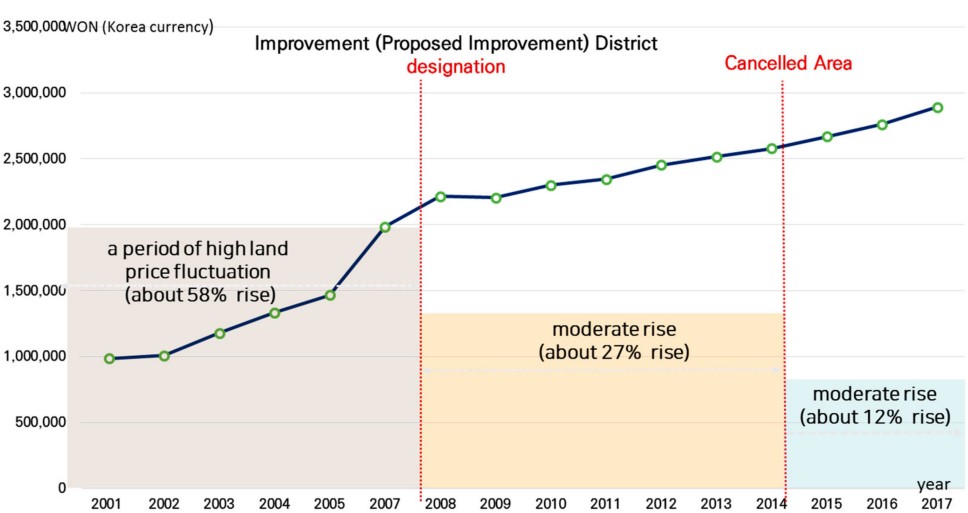

**Figure 11.** Changes in average land prices in the canceled area, 2001–2017 [11].

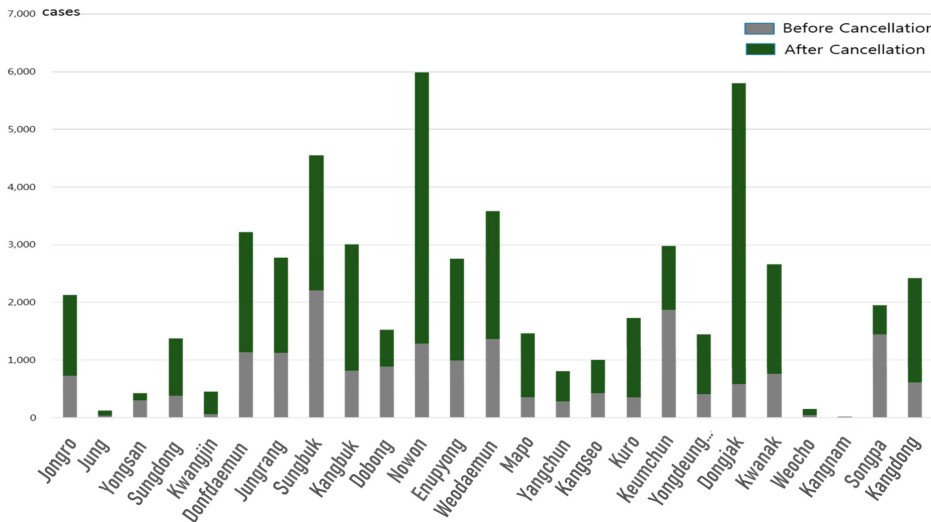

**Figure 12.** Case numbers of house sales before and after cancellation [11].

### 3.1.5. Sociocultural Characteristics

The aging index (ratio of elderly population over 65 years old to youth population 0–14 years old) is an indicator of the degree of aging of the population. The aging index of the canceled area was 239.2%, which was very high compared to the overall aging index of 97.2% in Seoul, indicating that the aging of the population was very serious (Figure 13). Among the houses in the canceled area, many had been vacant for more than 30 years. A hanok (traditional Korean house older than 50 years) is not just an old house, but a target of high value as a local asset. The public sector needs to play an active role in supporting and raising value awareness [11,12].

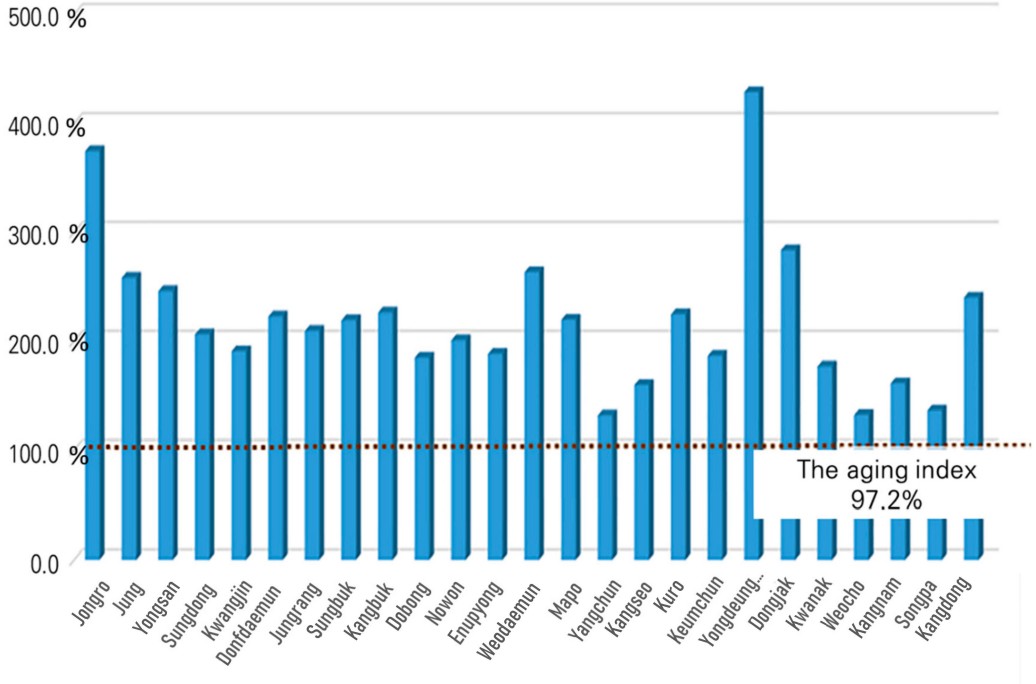

**Figure 13.** Aging index for canceled areas [11].

*3.2. Urban Regeneration Issues*

3.2.1. Physical Distribution Characteristics and Grouping

In this section, we examined urban regeneration issues in the released areas. We grouped similar objects with regard to the distribution of targets and release issues, focusing on actual cases of the lifted area.

As seen in Figure 14, one group corresponded to the targets that needed infrastructure maintenance, followed by the group that needed housing improvement. The targets that needed comprehensive maintenance were poor in both housing and infrastructure. This suggests that for most of the areas, the quality of the residential environment was lower due to infrastructure conditions rather than housing conditions. Therefore, it is necessary to encourage the improvement and installation of public infrastructure [11,12].

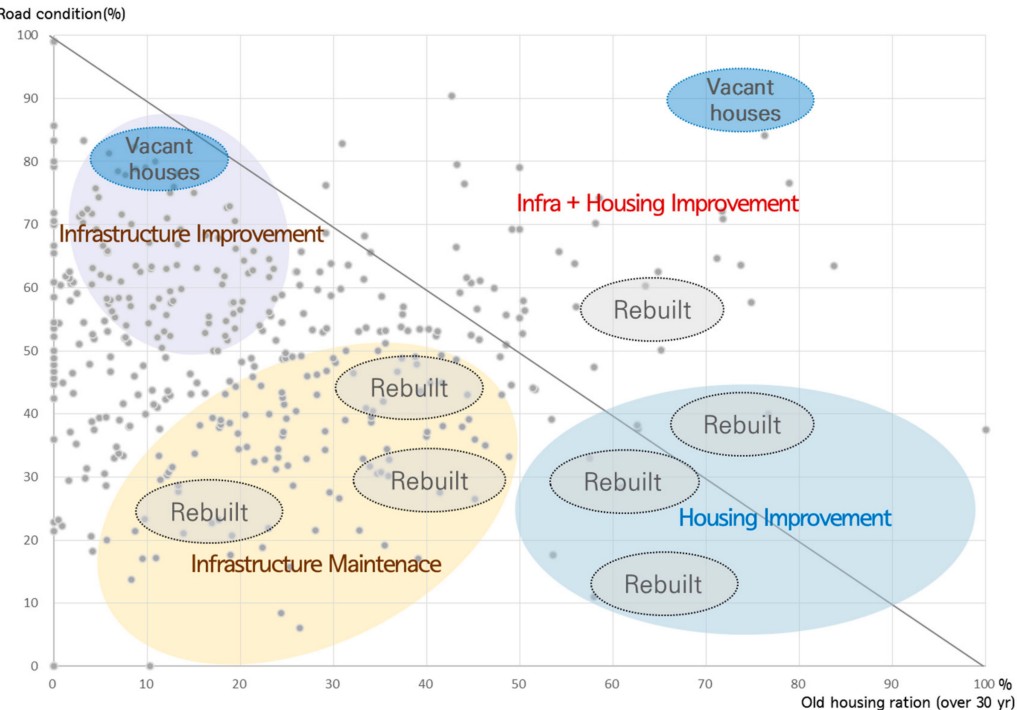

**Figure 14.** Distribution status and grouping [11].

3.2.2. Case Study

The area where comprehensive improvement was needed was the area where housing and infrastructure improvements were needed. Currently, the SMG's policy is difficult to implement regarding redevelopment of the area, so it is necessary for the public to implement housing improvement projects along with infrastructure improvement.

In this paper, we examined the regeneration issues that the released area should solve in the future, focusing on the case of Sajik 2 district. This area has experienced many serious urban problems since the cancellation. This study was intended to explain the urban problems of public policy failure through the case of Sajik 2 district.

Sajik 2 district was designated as a redevelopment project district on 19 November 2009. During the improvement project in 2017, the SMG forced the government to dismantle its authority under a big policy called "conservation of the historical resources of Seoul city wall." In April of the same year, after the dismissal of authority, the government started to establish a plan for the "Seoul-type urban regeneration project" in order to resolve the strong opposition and address residents' complaints. Since the release, land prices have risen sharply due to the expected effect of development profit due to

good locations in the city center. However, there are many vacant houses in the area (about 45% of the total buildings) and problematic areas, such as those prone to nighttime crime (Figure 15).

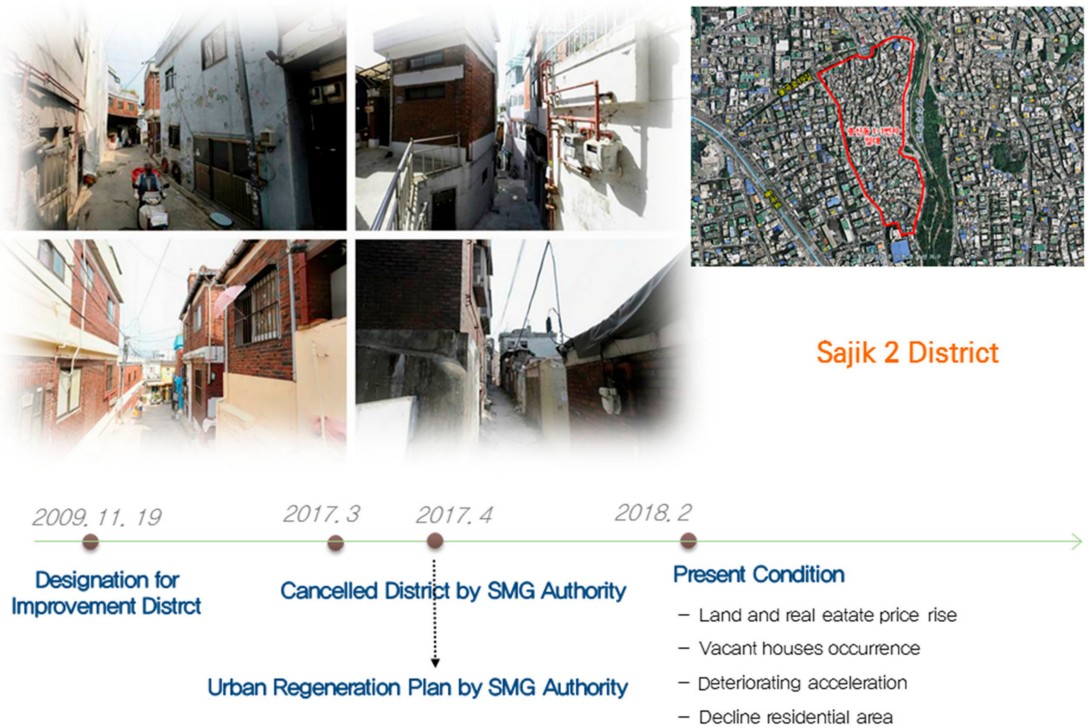

**Figure 15.** Present condition of Sajik 2 district.

Sajik 2 district is a low-rise residential area located in downtown Seoul. It is part of the old area of Gyeonghui Palace from the Joseon Dynasty. Various modern and contemporary buildings coexist outside this area. Characteristic historical resources inside the area include Western missionary houses, old waterways, and many hanoks (over 100 years old), which have strong historical identity, an uncommon feature in modern Seoul. Acting against the residents' wishes, the SMG carried out the forced release process, which turned this region into an area of great conflict due to strong distrust of and opposition to the Seoul city policy. Local residents were in a state of helplessness, not knowing what to do to improve the residential environment, such as improving old houses and installing new infrastructure.

The SMG began to establish a plan for residential environment management, with the planner currently reviewing various suggestions. About 33% of the buildings in this region are more than 30 years old, and about 20% of the total area consists of old houses that need renewal. There was not even a management plan in place for buildings over 50 years old. Sajik 2 district was canceled during the application process for redevelopment project approval, but local management and improvements are urgently needed, such as reducing the number of houses that were left vacant (Figures 16 and 17, Table 3). Local residents wanted a redevelopment project, but the SMG released the area to prevent high-rise construction and preserve the Seoul Castle, a historical resource.

**Table 3.** Present conditions of Sajik 2 district.

| Character | Detail |
| --- | --- |
| Area | 34,269 m$^2$ |
| Land use | Type 2 general residential area |
| Number of plots | 339 plots |
| Number of buildings | 220 buildings |

**Table 3.** *Cont.*

| Character | Detail |
|---|---|
| Housing density | 64% |
| Number of buildings more than 20 years old | 107 buildings |
| Ratio of buildings more than 20 years old | 49% |
| Number of buildings more than 30 years old | 72 buildings |
| Ratio of buildings more than 30 years old | 33% |
| Number of buildings more than 50 years old | 37 buildings |
| Ratio of buildings more than 50 years old | 17% |
| Number of plots facing roads over 4 m wide | 64 plots |
| Number of plots facing roads 3–4 m wide | 25 plots |
| Number of plots facing roads less than 3 m wide | 2 plots |
| Number of plots not in contact with a road | 9 plots |
| Land price increase rate (2017 compared to 2007) | 115% |
| Number of house sales | 113 |

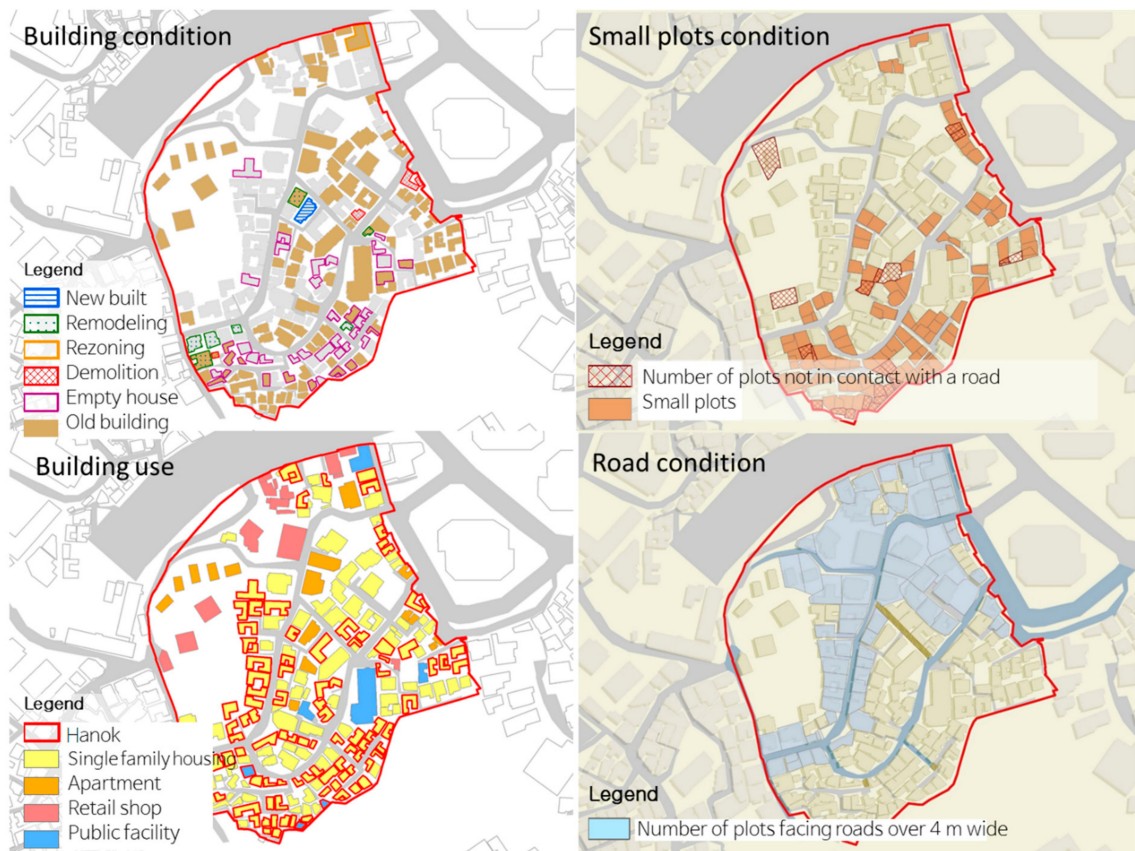

**Figure 16.** Present conditions map [11].

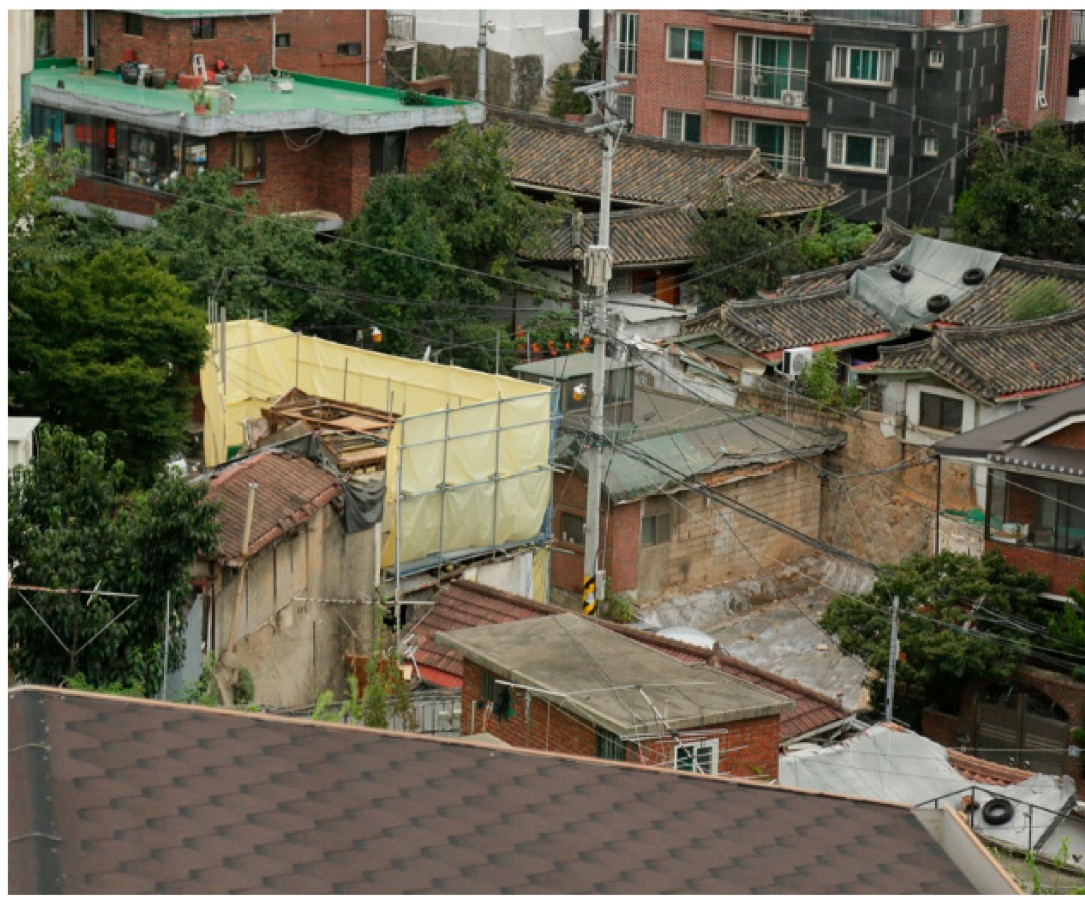

**Figure 17.** Picture of housing conditions.

Residents raised many complaints, including about what role the public should play with regard to area management, and questions about how they should fix their houses according to the suspension of the redevelopment project, how infrastructure such as parking and roads will be maintained, and how old, vacant houses in the crisis of collapse in the region will be handled (Table 4).

**Table 4.** Residents' complaints in Sajik 2 district.

| Complaint | Details |
|---|---|
| Parking and road-related | Secure and open roads to improve vehicle accessibility<br>Resolution of parking problems in the area |
| Historical preservation and hanoks | Treatment of the Korean houses that are left |
| New building act | Legal system of new housing construction |
| Public role and support | Refurbishment of public facilities<br>Deregulation plan to solve difficulties regarding new construction |

The regeneration issues regarding Sajik 2 district following its release can be put into several groups (Figure 18).

First, despite the strong desire of residents to carry out the project, distrust of the public (SMG) developed due to the forced release. In this process, a number of vacant houses were left in an old state; these have been grouped together. Houses with similar conditions were left vacant in succession and transformed into empty house areas; these empty houses are not worth using, but were still left as old, empty houses with high land prices. This presents a limitation in regional regeneration.

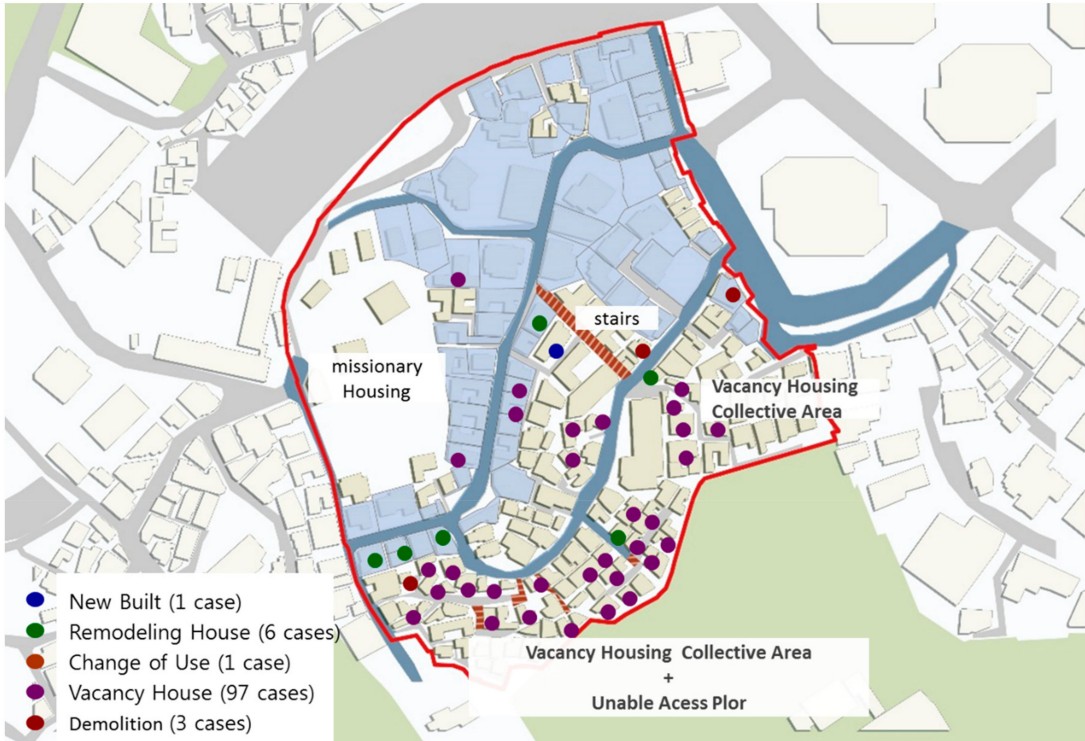

**Figure 18.** Present conditions map [11].

Second, there were many blind spots in the analysis of areas that are not in contact with any roads, and accessibility of these areas using stairs is very poor. However, there is still a problem in terms of public projects responsible for securing infrastructure being difficult due to the area's characteristics of irregular, scattered plots.

Third, housing located on blind properties and houses facing stairs have limitations when it comes to improvement. This is due to difficulty procuring construction materials and the poor working environment of laborers, because vehicles cannot be operated there, leading to difficulty in building construction.

Sajik 2 district has a very poor residential environment. The public sector should consider how to effectively solve the issue of regenerating the district, because it is a region where redevelopment of the entire demolition cannot be done according to current public policy decisions.

## 4. Conclusions

This work involved a basic study of the setting of policy direction using an analysis of characteristics of the SMG's policy failure from the viewpoint of urban regeneration using a case study.

The cancellation areas are an example of public policy failure. Urban regeneration that aims to improve small-scale alternatives is the best example of the perspective shift in this policy. Public responsibility is required with regard to canceling the unplanned affiliation policy and starting regeneration of the area [7]. Reversing failed policies can have a negative impact on real estate prices and the influx of local speculative capital, and requires enormous administrative power and finance.

In particular, there are various regeneration issues that need to be addressed, such as poor residential environments, speculation demand, and consideration for the socially underprivileged.

In this study, the distribution of improvement requirements of the area that was released was examined, with a focus on old and defective infrastructure, in order to understand the major regeneration issues; these areas were grouped and characterized. The most common issue found was the need to maintain infrastructure. This is due to the fact that the role of the public domain is very

broad, but it is necessary to involve the public in this plan for the active improvement and installation of public infrastructure.

This study dealt with more specific regional regeneration issues by focusing on case areas. According to the results, the public role in improving the physical environment of the canceled areas, as well as public distrust and conflict with residents (such as landowners) due to inappropriate public policy, can be further understood. Losing residents' trust in public policy is a serious problem that needs to be addressed in the future. Through this understanding, it is clear that efforts need to be made to reduce the deep worries of residents and increase publicity in establishing public policies, while also reflecting on them.

The Seoul Metropolitan Government needs to take responsibility for the failure of public policy through concrete strategies and measures for regeneration of the area.

First, it is necessary to prioritize public support and support for the response of urban problems in many of the canceled areas. Second, it is necessary to establish customized plans for various urban regeneration issues. Third, it is necessary to establish a comprehensive management plan for managing a wide range of canceled areas in the future.

**Funding:** This research received no external funding.

**Acknowledgments:** This study is a paper that revised and developed part of the policy project of the Seoul Research Institute in 2017, "Analysis of the actual conditions of the New Town and Redevelopment Area and the direction of residential regeneration."

**Conflicts of Interest:** The author declares no conflicts of interest.

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
