# Peer review of "A Study on the Characteristics of New Towns and the Redevelopment of Project-Canceled Areas: A Case Study of Seoul, South Korea"

_sustainability, doi:10.3390/su11205733_

Round 1

Reviewer 1 Report

First of all, congratulations for the great work done. Indeed, the manuscript present relevant issues. Nonetheless, still some minor reviews should be performed.

Before final publication, I would like to recommend to improve the sharpness of some figures.

In fig. 1 include the map legend.

In fig. 2 All text must go in English

In fig. 4 include the map legend.

The font is different in the following lines 172-179.

In fig. 10 improve text sharpness

In fig. 16 Improve visibility of map legends

In fig. 17 improve the map legend.

Author Response

Original Comments and Suggestions

In fig. 1 include the map legend.

In fig. 2 All text must go in English

In fig. 4 include the map legend.

The font is different in the following lines 172-179.

In fig. 10 improve text sharpness

In fig. 16 Improve visibility of map legends

In fig. 17 improve the map legend.

I did fix it all. I found need to improve the sharpness of figures, so I improned the sharpness and figures size.

I really appreciate your comments and efforts.

Thank you!!!!!! 

Reviewer 2 Report

Dear author,

The English language and style are fine but minor spell check required. Maybe pay attention to add a few more things on conclusion chapter.

Author Response

I found some of spell mistake, so I did fix. Thanks!!!

And also I made efforts for add more things on conclusion chapter, those are for suggestion of public policy and direction.

I really appreciate your comments and efforts.

Thank you!!!!!!

Reviewer 3 Report

The paper is entitled: A Study on the Characteristics of New Towns and the Redevelopment of Project-Canceled Areas: A Case Study of Seoul, South Korea.

The analysis focuses in the South Korean cities and especially in the city of Seoul explaining urban regeneration and public policy issues of several districts. A case study district is also presented.

My overall opinion on the paper is that despite the interesting subject there is a broad sense that several things are left unclear or that it is expected from the reader to make the necessary assumptions.

The paper misses a section where the structure of the research is analysed. I believe that the author should add a section or paragraph explaining what the reader should expect form this paper. For example  there is a case study presented in section 3.2.2 that is not mentioned before in the paper (except from the paper’s title). What is the purpose of describing the specific case study? Which is the additional information that this case study provides to the reader? Which are the additional conclusions connected to this case study?

Literature review is poor and oriented mainly to the urban regeneration literature about Seoul. In my opinion it should be far more extended in order to include also a review of urban regeneration issues in a broader sense referring also to other cases, cities, continents.

There is a broad literature about issues of urban regeneration and multiple approaches as well. I would suggest that the author makes a brief and explain her point of view and contribution on the current approaches / dialectic. This would help the reader understand better the research question of the paper and its contribution on the urban regeneration research at a national or international level. It would also help to understand what are the particularities of South Korean cities and especially of the city of Seoul against other cities in Asia or in other continents as well.

It would also be interesting that the author explains more how her approach about urban regeneration is connected to the urban sustainable development idea in order to make a better connection the journals subject area.

Please find also below several comments  that I believe could help improve this paper:

Line 67: I think a verb is missing here. “were ……  to as the “canceled areas”.

Line 74: improvement (of) management

Line 94: could you please explain the term sunset system? I am not aware of it.

Line 105: Could you please explain the meaning of achievability? Or define how was the project’s achievability measured?

Line 106:…without the propellant. Please explain. I am not sure I understand the meaning of propellant here.

Figure 1: There is no legend in the map and it is impossible to understand what the colored dots represent. In addition the three circles in the left of the map should have larger letter font as the content is not readable. I think that the three circles should have the same size or have a relevant scale to the numbers written inside. I mean that the middle circle should be much smaller than the other to as 29 in much smaller than the other two numbers. Otherwise the circles should have the same size without being scale sensitive.

Figure 2: this diagram is also very small and not readable.

Figure 3: There is a spelling error in the “Residential Area 313 Km2

I am not sure if this figure is useful at all. I think it does not add any information not already mentioned.

Figure 4: The map should have a legend.

Line 149: Could you define what is the Seoul-type regeneration model? What are its main characteristics/ targets etc?

Line 154: Could you please explain this better? I do not understand why did residents felt distrust toward the SMG.

Line 177-179: Please mind the syntax.

Line 234: This sentence is somehow confusing. Please avoid long sentences and rephrase. Do you mean that 36% of the housing that is older than 30 years old has a building ratio over 60%?

Figures 6 and 7: I am also not sure I understand what these histograms present. Could you please analyse it more in the text and also provide units in the vertical axis?

Line 262: …i.e. people did not expected not to see

Line 268: Perhaps you mean “As mentioned above, multi-family houses tended to be constructed in the canceled areas”.

Line 279-280: This sentence seems here irrelevant.  The author should consider analyzing further the traditional housing situation in Korea in order to support this argument.

 Figure 12: The horizontal axis is not readable.

Line 288-289: Please clarify the meaning of distributed targets.  In what way  / meaning are they distributed?

Figure 13: Please provide units in the vertical axis and explain how the road condition is defined. In the horizontal axis you probably mean old housing ratio instead of ration. Is this a percentage?

Line 298-299: I think there is a verb missing here. “…policy is difficult (to be implemented) regarding….”

Could you please explain why did you choose this case study? Is it a representative case?

Line 311: “…have begun to be exposed…” Please clarify what you mean with exposed. In what way? Or rephrase.

Line 320-321: Local residents were unable (to) gain any traction… Please clarify also the meaning of this sentence…you probably mean there were not appropriate incentives in order to motivate residents? Which were the residents’ wishes against SMG acted and created distrust?

Line 325-327: Could you please mention the reasons of “canceling” the district? Moreover it is not clear which were the exact  redevelopment actions that were planned and then canceled?

Table 3: Land price increase (since when?)

Figure 16: Legend is not readable.  Please also mention in the text what is the information presented in the four maps.

Lines 333-336: Please consider this suggestion: “…including what the role that the public (sector) should play……including questions around about how… ”

Table 4: Is the information on this table part of some research?  If this is the case please provide information about the way the research was conducted (for example questionnaire  etc..)

Line 340: Figure 1 reference seems irrelevant here.

Line 343: I do not understand how they have been grouped together. Perhaps the author refers to the information presented in fig 16 which is not readable? Moreover I do not understand why this is a limitation in regional regeneration. Could you be more precise?

Figure 17: Please consider adding a reference in the text.

Line 369: …focus on old elderly…in the area. Please rephrase properly. This sentence does not seem to make sense to me.

Line 370: …these areas were grouped and characterized.  This is also not clear to me… Maybe the author means the description of the characteristics of these areas?

Line 371: The most common area (issue) found… Maybe the word “issue” instead of “area” fits better here.

I have the impression that the conclusions are not supported well by the text. This is probably because the analysis leaves a lot of issues to be assumed by the reader. I am not sure I understood which is the exact issue of the failure of the canceled areas and how are the alternative regeneration plans structured and implemented. Which is their common basis?

Author Response

First of all, I really appreciate your comments and efforts.

I found many mistakes of my paper and tried to revise on your advice.

I fixed those from 1st line to last line on your comments.

   - add structure of the research section

   - conclusion revise

   - review spell and figures sharpness, etc,.

Thank you!!!!!!

Reviewer 4 Report

The author provides a case study of a failed building regeneration project in Seoul, Republic of Korea.

The description of the situation of the environment in the respective area of Seoul is quite informative and good. What I miss is a description of the legal background, for example if a district will be regenerated, what does that mean from the view of house owners? What kind of costs can the house owner expect? Or is the whole project financed by tax money? I ask that because in many Western countries a mayor can renovate roads or public space owned by the He legal background is important to understand the conflicts between house owners and city government which have evolved according the author the course of time.

What is a ‘union’? (on page 15 a few times)

Sometimes the English is flawed (only examples):

“Building ratio over 20 years”, does the author mean “the share of buildings older than 20 years”?  

“.. people expected to not see a profit for new houses through the redevelopment.”; maybe the author means, “the people did not expect to make a profit from new houses from the redevelopment.”?

Thus a careful proof-reading is strictly recommended; particularly regarding grammar and syntax mistakes.

Nearly all figures suffering from different deficiencies:

Fig 1:  the contrast of colors has to be improved.

Fig 2: What means “dis”

Fig 4: I have no clue for what that figure is good. What shall it illustrate?

Fig 6 & 7: What mean the numbers above the bars, what mean the percentage numbers, % of what?

Fig. 10, what are the units at the y-axis? Korean Won?

Fig 12: What is the aging index? Definition of it is needed.

Fig 13: What does 100 represent at the x and y axis?

Fig 16 & 17: What does the figure try to illustrate?  

Author Response

First of all, I really appreciate your comments and efforts.

I found mistakes of my paper and tried to revise on your advice.

I fixed those from 1st line to last line on your comments.

Thank you!!!!!!

Round 2

Reviewer 3 Report

The author has made an extensive improvement of her manuscript and the revised version is now better structured and presented. She has also provided all the necessary clarifications and details needed. 

I think that the paper still misses a broader analysis of the international literature but i find that the revised version can proceed for publication as the author has done her best to improve the overall presentation of the paper and the revised version is indeed very good.

Before providing the final version the author could consider revising figure 1, as I think that in the third circle of the diagram it should be more correct to write: canceled area condition analysis.

Author Response

Response to Reviewer 1 Comments

First of all, I really appreciate your kindly detailed review.

point 1 : I found grammar mistake figure 1.

Response 1

I’ve corrected to write : canceled area condition analyze canceled area condition analysis.(see figure 1, p3)

Thanks.

Reviewer 4 Report

The paper is much improved, and only a few adjustments have to be made.

-Please add a short information in brackets after “Cheonggyecheon” because I think otherwise most readers are not aware what it is.

-The new (first) paragraph in section 1.3 is written in past tense, it should be present tense.

In figure 1, it is written “Canceled area condition analyze”, I think the author means “condition analysis”.

Author Response

Response to Reviewer 2 Comments

First of all, I really appreciate your kindly detailed review.

point 1 : I absolutely agree with you, Cheonggyecheon is area in Seoul that needs additional explanation.

Response 1

I added and additional description to the Cheonggecheon. (see line 48 to 52, p2)

(the area is a 10.9-kilometre-long (6.8 mi), modern public recreation space in downtown Seoul. The massive urban renewal project is on the site of a stream that flowed before the rapid post-war economic development caused it to be covered by transportation infrastructure. The $900 million project initially attracted much public criticism but, since opening in 2005, has become popular among residents and tourists)

point 2 : I found tense mistake of section 1.3. It is written in past tense.

Response 2

I changed section 1.3to the present tense.(see line 93 to 103, pp2 ~ 3)

point 3 : I found grammar mistake figure 1.

Response 3

I’ve corrected to write : canceled area condition analyze canceled area condition analysis.(see figure 1, p3)
